# Pull-push neuromodulation of cortical plasticity enables rapid bi-directional shifts in ocular dominance

**Su Z Hong[1], Shiyong Huang[1†], Daniel Severin[1], Alfredo Kirkwood[1,2]\***

[1]Mind/Brain Institute, Johns Hopkins University, Baltimore, United States;
[2]Department of Neuroscience, Johns Hopkins University, Baltimore, United States

**Abstract** Neuromodulatory systems are essential for remodeling glutamatergic connectivity during experience-dependent cortical plasticity. This permissive/enabling function of neuromodulators has been associated with their capacity to facilitate the induction of Hebbian forms of long-term potentiation (LTP) and depression (LTD) by affecting cellular and network excitability. In vitro studies indicate that neuromodulators also affect the expression of Hebbian plasticity in a pull-push manner: receptors coupled to the G-protein Gs promote the expression of LTP at the expense of LTD, and Gq-coupled receptors promote LTD at the expense of LTP. Here we show that pull-push mechanisms can be recruited in vivo by pairing brief monocular stimulation with pharmacological or chemogenetical activation of Gs- or Gq-coupled receptors to respectively enhance or reduce neuronal responses in primary visual cortex. These changes were stable, inducible in adults after the termination of the critical period for ocular dominance plasticity, and can rescue deficits induced by prolonged monocular deprivation.

**\*For correspondence:**
kirkwood@jhu.edu

**Present address:** [†]Program in Neuroscience, Hussman Institute, Baltimore, United States

**Competing interests:** The authors declare that no competing interests exist.

## Introduction

Sensory experience can shape cortical circuitry and function. One clear demonstration of this is seen the visual cortex during a postnatal critical period, when brief visual deprivation in one eye can reduce the cortical responsiveness to the deprived eye while increasing the responsiveness to the fellow eye (for recent reviews see *Espinosa and Stryker, 2012*; *Hensch and Quinlan, 2018*). It has been widely recognized that this type of neural plasticity requires Hebbian-like forms of long-term potentiation (LTP) and depression (LTD) and neuromodulatory signals conveying information on the organism's behavioral state (for reviews see *Gu, 2002*; *Gu, 2003*; *Hofer et al., 2006*; *Kasamatsu, 1991*; *Smith et al., 2009*). How neuromodulators and Hebbian plasticity interact during cortical remodeling, however, is not fully understood.

Studies in vitro have demonstrated that neuromodulators can gate the induction of Hebbian plasticity by promoting excitability in cortical circuits (for reviews see *Brzosko et al., 2019*; *Foncelle et al., 2018*; *Frémaux and Gerstner, 2015*; *Palacios-Filardo and Mellor, 2019*; *Pawlak et al., 2010*). In addition, neuromodulators can also control the expression of synaptic plasticity, affecting its gain and polarity. These effects on the expression of synaptic plasticity result from the activation of distinct G-protein coupled receptors (GPCRs). A picture emerging from studies in the visual cortex is a simple rule in which Gs-coupled receptors that activate the adenylyl cyclase pathway and Gq-coupled receptors that activate the phospholipase-C (PLC) pathway have opposite effects on LTP and LTD (*Huang et al., 2010*; *Seol et al., 2007*). Gs-coupled promote LTP and suppress LTD, whereas Gq-coupled receptors promote LTD and suppress LTP. An attractive consequence of the pull-push rule is the wide range of possible outcomes, from and LTP-only state when activity of Gs dominates, to an LTD-only state when Gq dominates (*Huang et al., 2010*; *Huang et al., 2012*; *Huang et al., 2013*). Thus, the pull-push neuromodulation of LTP/D by the

balance of Gs/Gq activity could serve as a metaplasticity mechanism allowing a rapid reconfiguration of the plastic state of cortical synapses.

Motivated by the considerations above we examined whether GPCRs can shape experience-dependent plasticity in mouse V1 in vivo via a pull-push mechanism. Our results show that targeting receptors coupled to Gs or Gq in conjunction with visual stimulation can increase or decrease cortical response in a manner consistent with the pull-push model. These cortical changes were robust, stable in the absence of further visual conditioning, could be induced in adults long after the termination of the critical period for ocular dominance plasticity and can reverse ocular dominance imbalance induced by prolonged monocular deprivation.

## Results

We evaluated the pull-push neuromodulation of plasticity in vivo in the mouse visual cortex. We asked whether combining monocular stimulation or deprivation with the manipulation of GPCRs affects visual cortical responses in a manner predicted by the pull-push model, that is Gq-coupled receptors promoting depression and Gs-coupled receptors promoting potentiation. Changes in visual responses were monitored with optical imaging of intrinsic signal in V1 (*Kalatsky and Stryker, 2003*), and GPCRs were specifically targeted with pharmacology and chemogenetics (see Materials and methods).

### Systemic administration of GPCR agonists in conjunction with brief monocular visual conditioning induces bidirectional change of ocular dominance

We first tested whether brief monocular stimulation paired with activation of 5HT2c serotonergic receptors (paired to Gq) induces lasting depression of the visual responses, and whether the same visual stimulation, but paired with the activation of β-adrenergic receptors (βAR, coupled to Gs) potentiates the visual responses in juvenile mice (4 to 5 week old). We choose these particular receptors because in monocular deprivation, the disconnection of the deprived eye requires serotonin, while the potentiation of the fellow eye requires norepinephrine (*Nakadate et al., 2013*). In addition, βAR and 5HT2c are unique in supporting LTP and LTD, respectively, evoked with reinforcement like approaches (*He et al., 2015*).

The experimental schedule is depicted in *Figure 1A*. On the first day the cortical responses to either eye was measured in the binocular segment of V1 in one hemisphere. Next day the dominant eye (contralateral to the imaged hemisphere) was conditioned under isoflurane anesthesia with visual stimulation (1 hr. See Materials and methods for more details) in conjunction with systemic injection of one specific agonists (βAR: isoproterenol, 15 mg/kg, i.p.; 5-HT2c: Ro 60–0175, 10 mg/kg, i.p.). Following the visual conditioning, the mice were immediately brought into the dark to prevent visual experience until a later imaging session the following day.

In naïve mice the contralateral eye dominates the visual cortical response in the binocular zone, and the computed ocular dominance index is typically about 0.3 (see Materials and methods). In a first round of experiments we conditioned the contralateral eye in the presence of Ro 60–0175, the agonist for the Gq-coupled receptor 5-HT$_{2C}$. As shown in *Figure 1B,C* and in accordance with the role of Gq-coupled receptors in synaptic LTD, this conditioning specifically depressed the contralateral responses (before: 2.38 ± 0.23, after: 1.84 ± 0.21, p=0.037) without affecting the ipsilateral responses (before: 1.29 ± 0.11, after: 1.27 ± 0.13, p=0.75, n = 10), which translated in a significant decrease in the ODI magnitude (ODI, before: 0.28 ± 0.02, after: 0.14 ± 0.02, p=0.002, n = 10). In a similar fashion, visual conditioning paired with injection of an agonist for another Gq-coupled receptor (methoxamine:15 mg/kg, i.p., the α1-adrenergic receptor agonist), also lowered the ODI magnitude compared to those of the vehicle control group (ODI, methoxamine: 0.10 ± 0.04, n = 5; vehicle: 0.27 ± 0.02, n = 7, Wilcoxon rank sum test, p=0.001; *Figure 1—figure supplement 1B*). On the other hand, the same visual conditioning, but in the presence of isoproterenol, the agonist for Gs-coupled βAR, resulted in the opposite outcome in both the ODI and the response amplitude of the conditioned eye. The ODI increased after the visual conditioning, mainly by the potentiation of the conditioned eye, as the response amplitude from the other eye was not changed (response amplitude from conditioned eye, before: 2.74 ± 0.15, after: 3.52 ± 0.28, p=0.019; non-conditioned eye,

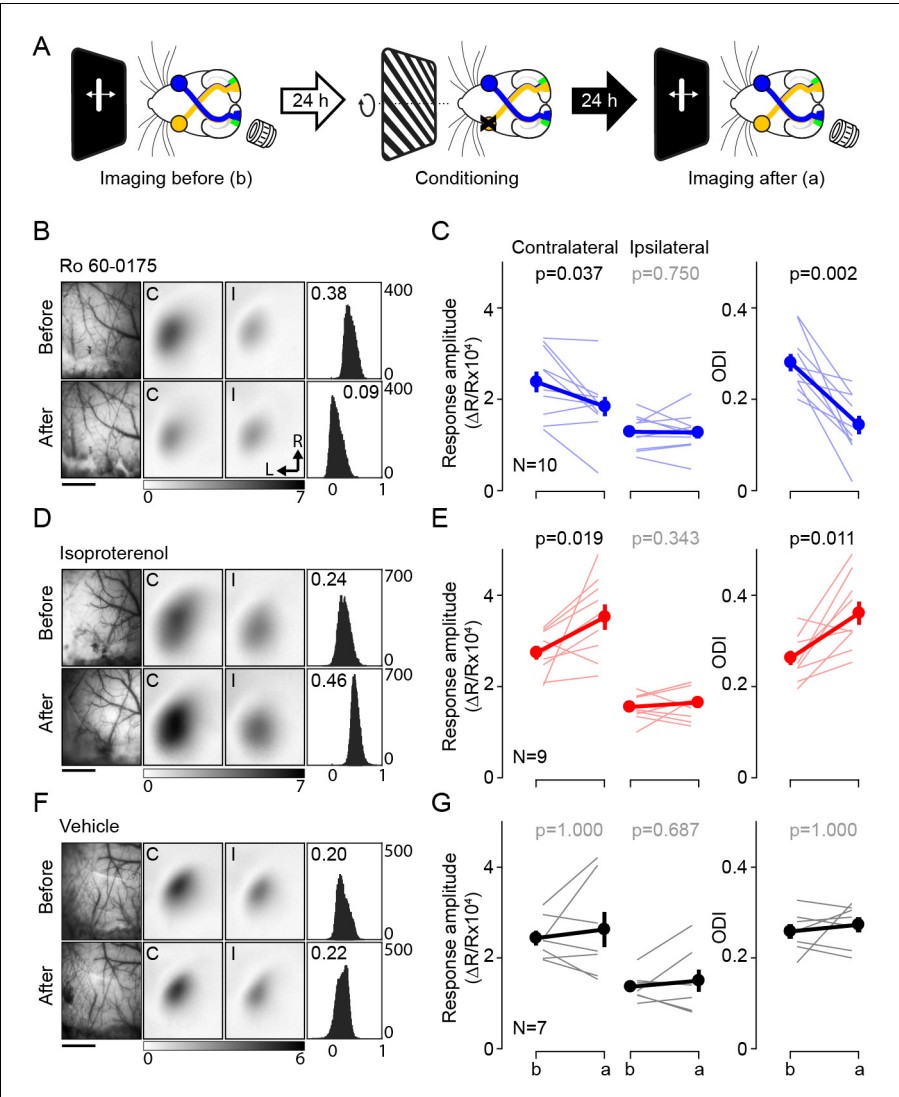

**Figure 1.** Pairing monocular stimulation with agonist for Gq- or Gs-receptors induce rapid and stable bidirectional changes of ocular dominance. (**A**) Experimental schematic. Left: The binocular segment of V1 in one hemisphere was imaged on day 1 to determine responses to ipsi and contralateral eye stimulation. Middle: 24 hr later the contralateral eye was stimulated with high contrast drifting gratings of 16 orientations for 1 hr starting 30 min after the administration of neuromodulator agonist (isoproterenol, 15 mg/kg, i.p.; Ro 60–0175, 10 mg/kg, i.p). Right: after 24 hr in the dark V1 was imaged again. (**B,C**) Pairing with the 5HT2c agonist Ro 60–0175 specifically reduces the response to the stimulated eye and reduces the contralateral bias. (**B**) Example experiment. Left: vasculature pattern of the imaged region used for alignment. Scale bar, 1 mm. Middle: magnitude map of the visual response from the eye contralateral [C] or ipsilateral [I] to the imaged hemisphere. Gray scale (bottom): response magnitude as fractional change in reflection x10$^4$. Arrows: L, lateral, R, rostral. Right: histogram of ocular dominance index (ODI) in illustrated in the number of pixels (x-axis: ODI, y-axis: number of pixels) (**C**) Summary of changes in response amplitude of the conditioned contralateral eye (left) and the non-conditioned ipsilateral eye (middle) as well as the change of ODI (right) before (**b**) and after (**a**) conditioning. Thin line: individual experiments; thick line and symbols: average ± s.e.m. (**D,E**) Pairing with the β-adrenergic agonist isoproterenol enhances the response to the stimulated eye and the contralateral bias. Example experiment in D, summary results in E, conventions as in B, C. (**E,F**) Visual stimulation alone does not affect either the visual response or the contralateral bias. Example experiment in E, summary results in F, conventions as in B,C.

The online version of this article includes the following source data and figure supplement(s) for figure 1:

**Source data 1.**
**Figure supplement 1.** Comparison of ODIs in various conditioning conditions.
**Figure supplement 1—source data 1.**

before: 1.55 ± 0.09, after: 1.65 ± 0.12, p=0.344; ODI, before: 0.26 ± 0.02, after: 0.36 ± 0.03, p=0.011, n = 9; *Figure 1D and E*).

As controls, we confirmed that the monocular visual conditioning alone did not induce significant changes in the ODI as well as the response amplitude (response amplitude from conditioned eye, before: 2.43 ± 0.16, after: 2.62 ± 0.39, p=1; from non-conditioned eye, before: 1.36 ± 0.11, after: 1.50 ± 0.24, p=0.687; ODI, before: 0.26 ± 0.02, after: 0.27 ± 0.02, p=1; n = 7) (*Figure 1F and G*). In addition, the agonists alone (without any visual stimulus) did not affect the responses and the ODI (ODI; naïve: 0.26 ± 0.01, n = 26; isoproterenol: 0.27 ± 0.02, n = 7; Ro 60–0175: 0.26 ± 0.03, n = 6; methoxamine: 0.26 ± 0.01, n = 6; $F_{3,41}$ = 0.09, p=0.964) (*Figure 1—figure supplement 1*). Altogether the results indicate that activation of GPCR signals can promote rapid and persistent bidirectional changes in ocular dominance. Importantly, the polarity of the changes is consistent with the pull-push model, with Gq-coupled receptors promoting depression, and Gs-coupled receptors promoting potentiation. Also, worth noting, only 1 hr of conditioning with neuromodulator induced changes in ODI of comparable magnitude of those induced by several days of monocular deprivation (*Cang et al., 2005*; *Sato and Stryker, 2008*).

## Activation of gq- or Gs-coupled DREADD promotes pull-push neuromodulation

The systemic administration of agonists can stimulate receptors outside the visual cortex, and potentially outside the nervous system, complicating the interpretation of the results. To confirm the role of the cortex in the changes described above, we employed a chemogenetic approach and virally transfected the visual cortex to express designer receptors exclusively activated by designer drugs (DREADDs). Thus, the systemic administration of the DREADDs agonist will only affect the targeted region. In the case of Gq-DREADDs we injected an adeno-associated virus (AAVs) containing neuronal DREADDs gene; in the case of Gs-DREADDs we used Gs-DREADDs knock-in mice (*Akhmedov et al., 2017*) injected with AAVs containing pan-neuronal Cre recombinase gene (rAAV-hSyn-Cre) (see Materials and methods for more details).

First, we examined if the Gq- and Gs-DREADDs can promote LTD and LTP in a pull-push manner as the endogenous GPCRs do (*Huang et al., 2012*; *Seol et al., 2007*). The assessment was done in visual cortical slices, in layer 2/3 pyramidal cells expressing Gq- and Gs-DREADDs. We used a spike-timing dependent plasticity (STDP) protocol (*Figure 2—figure supplement 1A and B*; see Materials and methods for details) because in our standard experimental conditions this paradigm depends crucially of the addition of GPCR agonists. The STDP protocol was applied to two pathways, P1 and P2: in P1 the stimulation preceded a burst of postsynaptic action potentials and in P2 it occurred after. This STDP pairing was delivered at the end of a 10 min bath application of the agonist clozapine N-oxide (CNO, 10 µM). Consistent with the pull-push model and with previous results (*Seol et al., 2007*), this conditioning (CNO and STDP) induced a robust LTD in both pathways in cells expressing Gq-DREADDs (P1: 60.0 ± 7.4%, p=0.031, n = 6; P2: 65.5 ± 7.3%, p=0.031, n = 6), robust LTP in cells expressing Gs-DREADDs (P1: 123.5 ± 3.4%, p<0.001, n = 9; P2: 121.2 ± 5.4%, p=0.045, n = 8), and has no effect in Gq- or Gs-DREADDs negative cells (Gq-DREADDs, P1: 96.6 ± 1.2%, p=0.125, n = 4; P2: 101.8 ± 3.8%, p=0.063, n = 5; Gs-DREADD, P1: 89.8 ± 9.9%, p=0.125, n = 4; P2: 84.0 ± 4.1%, p=0.250, n = 3).

Altogether, the results above validated the use of DREADDs. Therefore, to ascertain the role of cortex in the pull-push neuromodulation of ODP, we tested if the systemic injection of CNO (10 mg/kg, i.p.) in conjunction with monocular stimulation can promote bidirectional change of ocular dominance in mice expressing Gq- or Gs-DREADDs in vivo. As depicted in *Figure 2A*, the experimental schedule was very similar to the one use for the agonist for endogenous GPCR. The results were similar to that obtained with pharmacology. Monocular stimulation of mice expressing Gq-DREADDS in V1 in conjunction with systemic injection of CNO significantly reduced the ODI in the contralateral binocular cortex (ODI, before: 0.21 ± 0.01, after: 0.06 ± 0.03, p=0.031, n = 6; *Figure 2B and C*). The injection of CNO alone in the dark did not change the ODI (ODI, before: 0.32 ± 0.04, after: 0.31 ± 0.03, p=0.875, n = 4). On the other hand, in the mice expressing the Gs-DREADDs monocular conditioning caused a lasting and significant increase in the responses of the conditioned eye (response amplitude from conditioned eye, before: 2.89 ± 0.25, after: 3.41 ± 0.36, p=0.042; non-conditioned eye, before: 1.73 ± 0.14, after: 1.80 ± 0.16, p=0.845) and an increase in the ODI (ODI, before: 0.24 ± 0.02, after: 0.30 ± 0.02, p=0.018, n = 11; *Figure 2D and E*). A subset of experimental

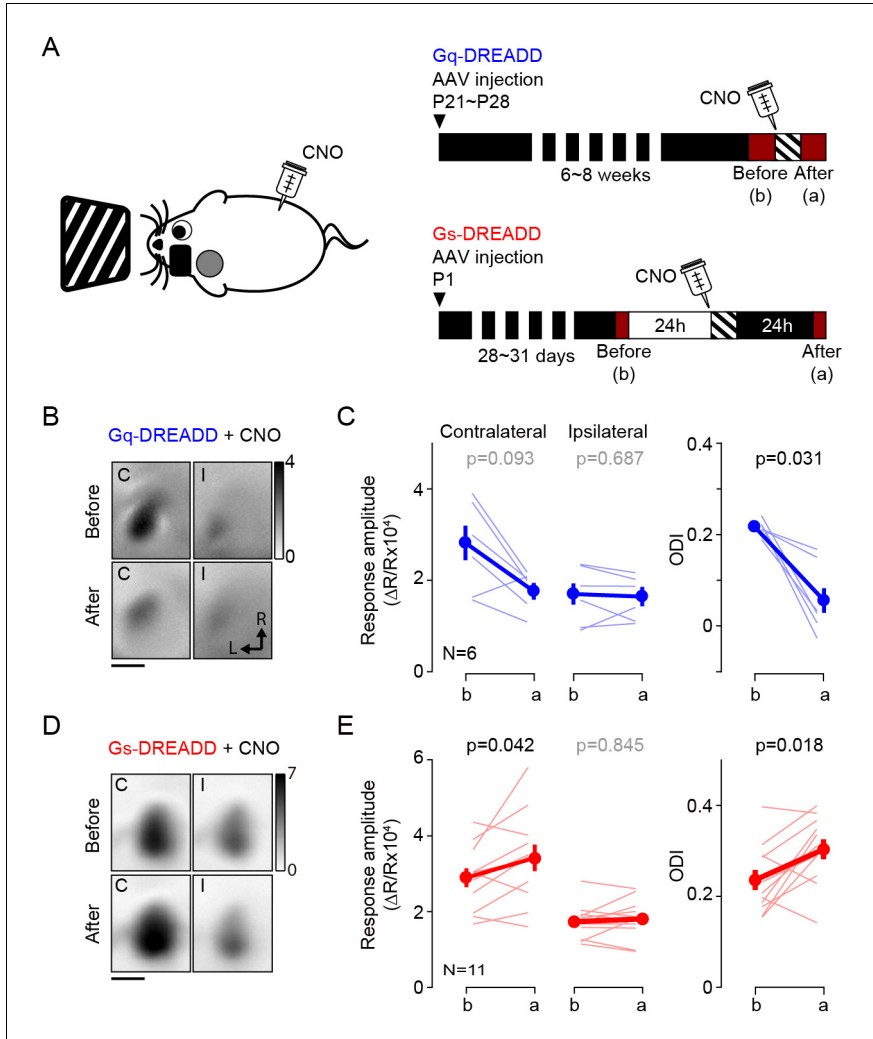

**Figure 2.** Activation of Gq or Gs-DREADDs enables rapid bidirectional changes of ocular dominance with visual stimulation. (**A**) The schematics of the conditioning. One hour of monocular stimulation 30 min after CNO injection (10 mg/kg) is shown on the left; the experimental time schedule for the viral injections, conditioning and optical imaging is on the right. For Gq-DREADDs (upper right) AAVs encoding neuronal Gq-DREADDs was injected into V1 of C57BL/6J mice at P21-28, and 4–6 weeks later visual responses were imaged before and after conditioning. For Gs-DREADDs (lower right) AAVs encoding hSyn-Cre were injected at postnatal day one into the posterior region of left hemisphere of Cre-dependant Gs-DREADDs mice, and visual responses were imaged between P28 to P33, a day before and after the conditioning. (**B,C**) In Gq-DREADDs expressing mice pairing visual stimulation with the agonist CNO selectively depresses the responses to the stimulated eye and reduces the ODI. An example experiment showing visual responses imaged before (upper panels) and after (lower panels) conditioning with activation is presented in **B**). Each panel represents the magnitude map of the visual response from the eye contralateral (left) or ipsilateral (right) to the imaged hemisphere are illustrated. Gray scale (upper right) shows response magnitude as fractional change in reflection x10$^4$. Scale bar, 1 mm. Arrows: L, lateral, R, rostral. (**C**) Summary of changes in response amplitude of the conditioned contralateral eye (left) and the non-conditioned ipsilateral eye (middle) as well as the change of ODI (right) before (**b**) and after (**a**) conditioning. Thin line: individual experiments; thick line and symbols: average ± s.e.m. (**D,E**) In Gs-DREADDs expressing mice pairing visual stimulation with the agonist CNO selectively potentiates the responses the responses to the stimulated eye and increases the ODI. Example experiment in **D**, summary results in **E**, conventions in B,C.

The online version of this article includes the following source data and figure supplement(s) for figure 2:

**Source data 1.**

**Figure supplement 1.** Pull-push regulation of LTP and LTD by activation of Gq- or Gs-coupled DREADD in L2/3 pyramidal neurons in V1.

**Figure supplement 1—source data 1.**

mice failed to express Gs-DREADDs in V1; in these cases, the ODI did not change with conditioning (ODI, before: 0.26 ± 0.11; after: 0.24 ± 0.13, p=1, n = 4). Although we cannot rule out that the different transfection approaches might have enhanced or moderated the opposite effects obtained with the Gq and Gs-DREADDs, the simplest explanation for the outcome is that the DDREADs affected plasticity as predicted by the pull-push idea. Altogether these results from the Gq- or Gs-DREADDs expressing mice support the idea that the direct activation of the GPCRs in V1 neurons is sufficient to drive the neuromodulation of ocular dominance plasticity.

## Intracortical pathways modified by pairing monocular stimulation with neuromodulators and by monocular deprivation

It was of interest to identify intracortical circuits within V1 affected by the neuromodulator pairing conditioning, and to compare them with the modifications induced by standard paradigms of experience dependent plasticity. Therefore, we evaluated in visual cortical slices the changes in postsynaptic strength induced by 1 hr conditioning in the presence of 5-HT2c agonist Ro 60–0175 and by 3 days of monocular deprivation (MD), as both manipulations result in a comparable decrease in the contralateral bias. The slices were prepared from the monocular V1 contralateral (MD, monocular deprived) or ipsilateral (ND, non-deprived) to the deprived eye in MD mice, and contralateral (C, conditioned) or ipsilateral (NC, non-conditioned) to the stimulated eye in mice conditioned with Ro 60–0175. Prior studies indicate that after p21 synaptic plasticity in V1 is restricted to non-granular layers, therefore we focused on inputs onto layer 2/3 pyramidal cells, specifically on the lateral inputs from other layer 2/3 cells and the ascending inputs from layer 4, both of which have been well studied in deprivation paradigms (*Maffei and Turrigiano, 2008*; *Petrus et al., 2015*). The experiments were done in slices from mice that express channelrhodopsin-2 (ChR2) in layer four neurons (*Petrus et al., 2015*), thus L4 inputs were stimulated optogenetically with blue light while L2/3 input were stimulated electrically (*Figure 3A*). For these two inputs we determined the quantal size (the postsynaptic response to the release of a single neurotransmitter vesicle) using the strontium ($Sr^{2+}$) to desynchronize the release of presynaptic neurotransmitter vesicles (*Bekkers and Clements, 1999*). Under these conditions the average quantal size (the response evoked by a single vesicle) can be directly obtained from the isolated events recorded shortly after stimulation (400 ms windows, *Figure 3B*), after correcting for spontaneous events occurring before the stimulation (*Petrus et al., 2015*; *Tran et al., 2019*; see Materials and methods for details). The results are reported in *Figure 3*. Compared to cells from non-deprived (ND) hemisphere, in cells from deprived (MD) hemisphere the average amplitude of the computed evoked events is reduced in both L4 inputs (ND: 15.9 ± 0.7 pA, n = 7 mice, 10 cells; MD: 13.2 ± 0.8 pA, n = 6 mice, 12 cells, p=0.011; *Figure 3C*) and L2/3 inputs (ND: 15.5 ± 0.9 pA, n = 5 mice, 9 cells; MD: 12.5 ± 0.7 pA, n = 6 mice, 8 cells, p=0.027; *Figure 3D*). On the other hand, in the hemisphere conditioned (C) with monocular stimulation combined with injection of serotonergic agonist Ro 60–0175 the average amplitude of the events evoked by stimulating layer four inputs was severely reduced compared to the average in non-conditioned (NC) hemisphere (NC: 19.9 ± 1.3 pA, n = 5 mice, 8 cells; C: 14.2 ± 1.0 pA, n = 6 mice, 11 cells, p=0.002; *Figure 3E*). In contrast, the average amplitude of the events evoked by layer 2/3 stimulation was not affected by the conditioning (NC: 16.5 ± 1.4 pA, n = 5 mice, 12 cells; C: 15.1 ± 1.1 pA, n = 6 mice, 11 cells, p=0.926; *Figure 3F*). These results indicate that ascending excitatory inputs from layer 4 cells onto pyramidal cells in layer 2/3 are depressed by both type of conditioning. Lateral inputs, however, were significantly reduced only by MD, possibly because the neuromodulators conditioning was done under anesthesia (to minimize complications due to endogenous neuromodulatory tone), which would reduce the intracortical activity required for Hebbian plasticity. Nevertheless, the results are consistent with the idea that conditioning with neuromodulators act on pathways that are also affected by monocular deprivation.

## Blockade of β-Adrenergic receptors during monocular deprivation induces ocular dominance shifts toward the visually deprived eye

According to the pull-push hypothesis of neuromodulation, the balance of the Gq/Gs-coupled receptor signals determines the polarity of the synaptic plasticity. This balance can be shifted to favor LTP or LTD not only by agonists for specific GPCRs but can also be altered with antagonists. Previously we showed that the pharmacological blockade of β-adrenergic receptors is sufficient to

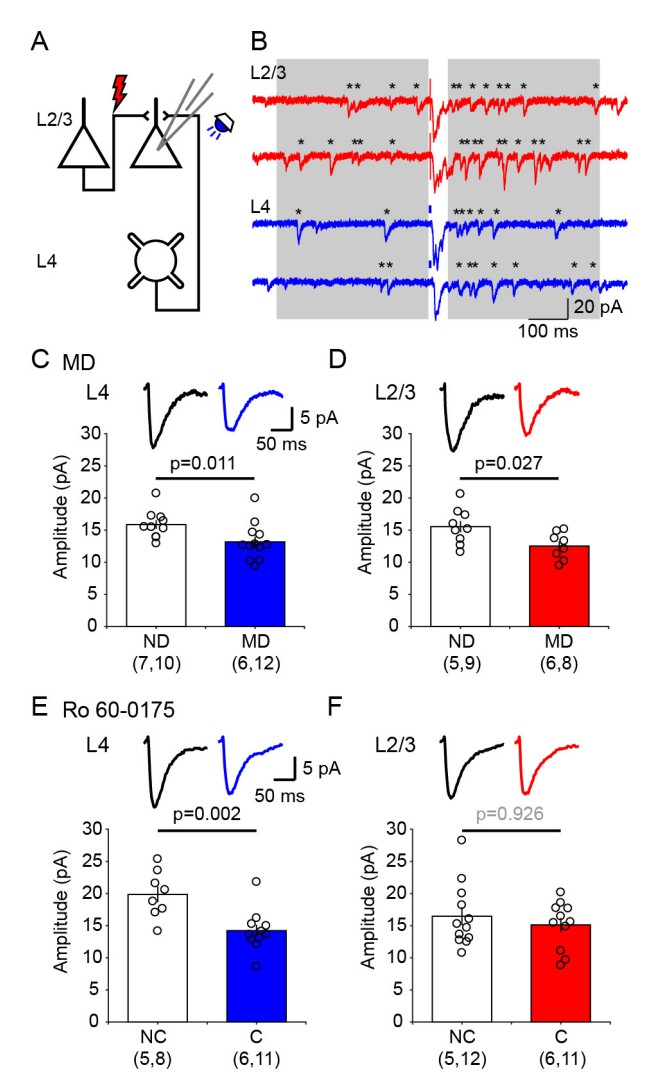

**Figure 3.** Ex vivo assessment of intracortical pathways modified by visual conditioning with Ro 60–0175 and by monocular deprivation (MD). (**A**) Experimental schematics. Synaptic responses were recorded in L2/3 pyramidal cells and evoked electrically with an electrode placed 100–200 μm laterally to activate L2/3 inputs, and optogenetically with whole field flash stimulation (5 ms, 470 nm or white LED) to activate L4 inputs. (**B**) Example experiment showing $Sr^{2+}$-induced desynchronization of responses to stimulation of L2/3 inputs (red traces) and L4 inputs (blue traces). Grey boxes indicate the time windows used to compute average evoked events. Individual events indicated with asterisks. (**C, D**) MD for 3 days reduced the average amplitude of quantal events evoked from both L4 (**C**) and L2/3 (**D**) inputs. The graphs show the computed average amplitude events in cells from MD mice and age-matched non-deprived mice cell. Traces on top are grand averages of events computed in all cells from all mice in the indicated condition. (**E, F**) Pairing monocular stimulation with Ro 60–0175 reduced the average amplitude of quantal events evoked from L4 (**E**) inputs, but not from L2/3 (**F**) inputs. Graphs and traces as in C, D. In parenthesis in (**C–F**) is the number of mice and cells. Experimenters were blind to the identity of the hemisphere recorded.

The online version of this article includes the following source data for figure 3:

**Source data 1.**

bring an LTD-only state in slices, and also in vivo (*Huang et al., 2012*). We asked therefore whether this intervention could also alter ocular dominance plasticity. We predicted that antagonizing β-adrenergic receptors during brief monocular deprivation (MD) would promote the depression of the responses to the most active eye: the non-deprived eye. This is exactly the opposite of the normal

outcome of brief MD: depression of the derived eye. In these experiments we sutured one eyelid for 2 days in juvenile mice (P28-31) for 2 days, and at the same time we infused the βAR antagonist propranolol systemically with an osmotic pump (5 mg/kg/day, s.c.). The changes in cortical responsiveness were monitored in the hemisphere ipsilateral to the deprived eye to maximize detecting a depression of the non-deprived eye. Note that this arrangement is opposite to the one used in standard monocular deprivation studies that monitor the contralateral cortex. As shown in *Figure 4B*, in the vehicle control group the 2 days of MD increased the ODI (before: $0.24 \pm 0.01$, after:

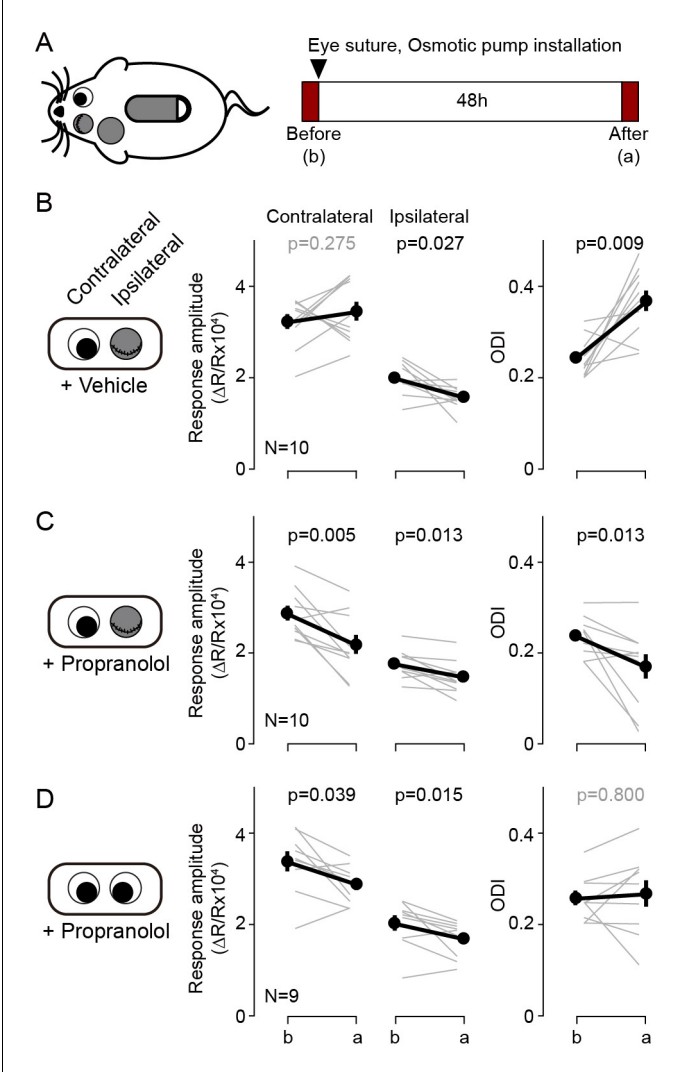

**Figure 4.** Blockade of β-adrenergic receptor during short-term monocular deprivation induces paradoxical depression of the open eye. (**A**) Experimental schematics. At P28-31, the eye ipsilateral to the imaged hemisphere was closed by eyelid suture for two days (short-term MD). An osmotic pump filled with β-adrenergic receptor antagonist propranolol was installed subcutaneously shortly before the eye suture. Visual responses from each eye were imaged immediately before and after the short-term MD. Note that deprivation and imaging are done in the same side of mice, not in opposite sides as in all other figures. (**B**) In mice infused with vehicle, MD induced a selective depression of responses to the deprived ipsilateral eye, no changes in the responses to the open contralateral eye and a concomitant increase in the ODI. (**C**) In mice infused with the propranolol, MD depressed both responses and decreased the ODI. (**D**) Propranolol infusion in non-deprived mice causes a modest depression of both ipsi- and contralateral responses, but without altering the ODI. Thin lines in B-D represent results from individual experiments, thick lines and symbols represent averages ± s.e.m.

The online version of this article includes the following source data for figure 4:

**Source data 1.**

0.37 ± 0.02, p=0.009, n = 10), largely due to a reduction of in the response to the deprived eye (before: 1.98 ± 0.11, after: 1.56 ± 0.08, p=0.027), and without changes in responses to the non-deprived eye (before: 3.22 ± 0.17, after: 3.44 ± 0.22, p=0.275) (*Figure 4B*). In contrast, in the pro-pranolol-treated group MD decreased the ODI (before: 0.24 ± 0.01, after: 0.17 ± 0.03, p=0.013, n = 10) and reduced the responses to both the non-deprived eye (before: 2.85 ± 0.17, after: 2.15 ± 0.22, p=0.005) and the deprived eye (before: 1.74 ± 0.10, after: 1.48 ± 0.10, p=0.013). On other hand the propranolol treatment itself did not affect the ODI in non-deprived mice (before: 0.25 ± 0.01, after: 0.26 ± 0.03, p=0.800, n = 9. *Figure 4D*). The results indicate that manipulations that alter Gs/Gq-coupled signals can affect the outcome of MD in a manner predicted by the pull-push model. According to the model, the paradoxical shift in ocular dominance towards the deprived eye promoted by blocking Gs-coupled β-adrenergic receptors will be the consequence of a predominance of Gq-coupled signaling favoring an LTD-only state in the cortex. In that scenario, the most active the inputs, the one from the non-deprived eye, will undergo more LTD.

### Systemic administration of GPCR agonists in conjunction with brief monocular visual conditioning induces bidirectional change of ocular dominance after critical period

Experience dependent cortical plasticity diminishes as the animal mature (*Heimel et al., 2007*; *Lehmann and Löwel, 2008*; *Pizzorusso et al., 2002*; *Sato and Stryker, 2008*; *Sawtell et al., 2003*). In juvenile mice MD induces depression of the responses evoked by the occluded eye and potentia-tion of responses to the open eye, but only during at the early stages of the critical period. MD does not depress the responses to the occluded eye in mice older than ~p35, and it does not potentiate the responses to the open eye in mice older than p90 (*Lehmann and Löwel, 2008*; *Sato and Stryker, 2008*; *Sawtell et al., 2003*). It was of interest, therefore, to examine whether stimulation of GPCRs can promote bidirectional changes in ocular dominance after the end of the critical period. We first tested in adult mice (P130-140) the effects of monocular visual conditioning (1 hr) in conjunc-tion with systemic injection of the α1 adrenergic agonist (coupled to Gq) methoxamine (15 mg/kg, i. p.), which effectively reduced ODI in juveniles (*Figure 1—figure supplement 1B*). Like in juveniles, in the adults the conditioning induced a robust and significant decrease in the ODI (ODI, before: 0.30 ± 0.02, after: 0.15 ± 0.03, p=0.003, n = 9; *Figure 5A*).

Next, we tried the complementary manipulation, to enhance the contralateral bias in adults by pairing the same visual conditioning with the β-adrenergic agonist (coupled to Gs) isoproterenol (15 mg/kg, i.p.). Unlike what we previously observed in juveniles (see *Figure 1A*), the pairing with iso-proterenol did not affect the ODI in adults (ODI, before: 0.31 ± 0.02, after: 0.32 ± 0.01, p=0.5, n = 4; *Figure 5B*). We reasoned that one factor that could have obscured the outcome is the age-depen-dent increase in the contralateral bias (*Lehmann and Löwel, 2008*). Indeed, in naïve adult mice (>P130) the ODI is significantly higher in the juvenile mice (p28 ~p35) and comparable to the ODI value in juveniles after monocular visual conditioning with isoproterenol (ODI, juvenile: 0.26 ± 0.01, n = 43; old: 0.30 ± 0.01, n = 18; juvenile after Iso: 0.35 ± 0.01, n = 9; $F_{2,67}$ = 9.59, p=0.008; *post hoc* Holm-Šídák test p=0.029; *Figure 5C*). These results suggest a ceiling effect. The contralateral bias of the visual response might be already saturated in adult mice, thus occluding further increases in ODI.

We examined possible ceiling effects in two ways. In one set of studies, we first preconditioned eight mice with methoxamine to 'unsaturate' the ODI, then after confirming 24 hr later the reduction in ODI, we tested the effects of isoproterenol conditioning (*Figure 5D*). As shown in *Figure 5E*, and consistent with the ceiling hypothesis, preconditioning reduced the ODI from 0.322 ± 0.018 to 0.136 ± 0.018, and conditioning with isoproterenol increased the ODI to 0.200 ± 0.024. A repeated measures ANOVA followed by a Bonferroni's test confirmed the significance of the differences (n = 8, $F_{6,12}$ = 3.238, p=0.039; *post hoc* Bonferroni's test, Before-After Mtx, p=0.002; After Mtx-After Iso, p=0.027). In another set of studies, we performed the isoproterenol-conditioning in mice subjected to long-term monocular deprivation (LTMD). This deprivation paradigm induces a pro-found reduction of the contralateral bias that is not fully reversed by reopening the deprived eye (*He et al., 2007*; *Pizzorusso et al., 2006*; *Sale et al., 2007*). In these experiments we closed one eye at p21 and reopened it after p130. Cortical responses were imaged in the cortex contralateral to the deprived eye as shown in *Figure 5F*, first shortly after eye reopening and 3 days later, to assess spontaneous recovery. Then the conditioning with isoproterenol was performed and the

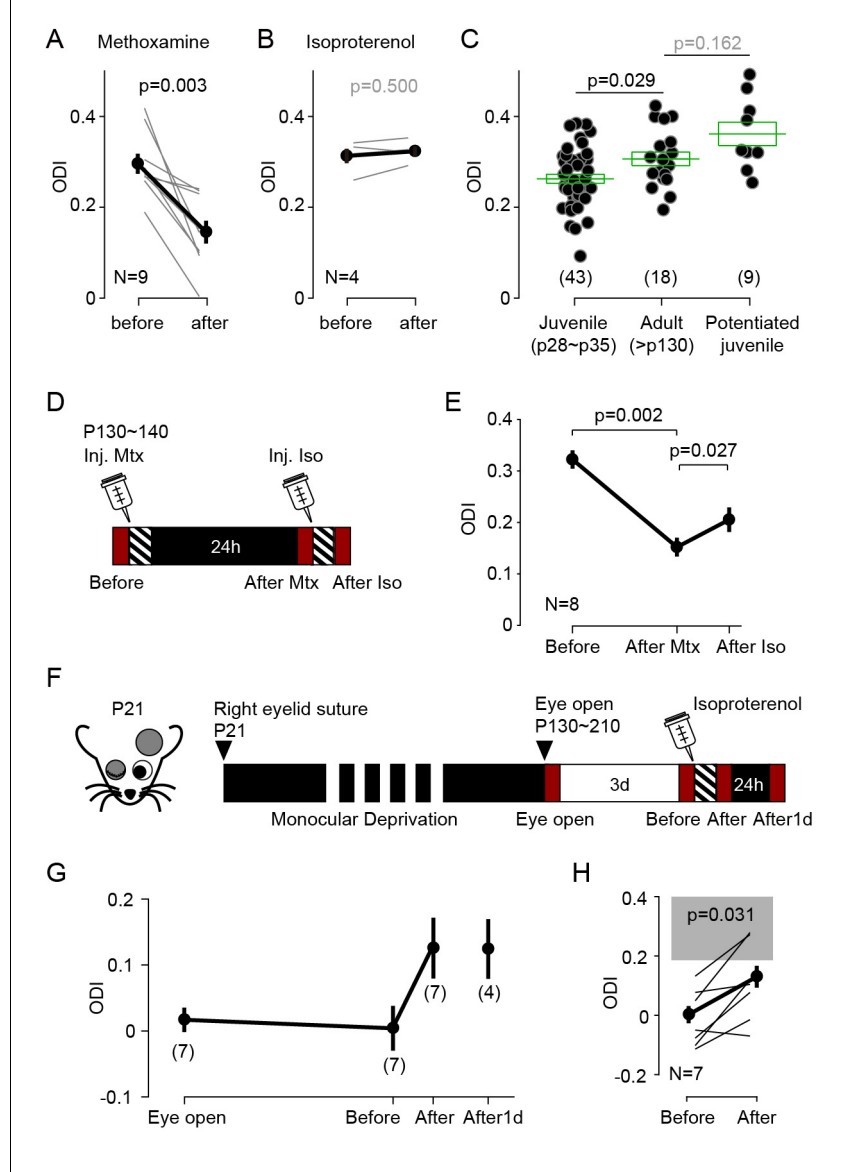

**Figure 5.** Neuromodulator agonists enable rapid bidirectional changes of ocular dominance after critical period. (**A**) In adult mice (>P130) pairing monocular stimulation with systemic injection of the α-adrenergic receptor (Gq-coupled) agonist methoxamine induced a robust decrease of ODI. (**B**) The same conditioning, but paired with the β-adrenergic agonist isoproterenol, failed to change the ODI in adult mice. (**C**) In naïve adult mice ODI values are larger than ODI values of naïve juvenile mice and comparable to the ODI values in isoproterenol-enabled potentiation juvenile mice (One-way ANOVA, $F_{2,67}$ = 7.043, p<0.01; *Post hoc* Holm-Sidak t-test). Filled circles represent data from individual mice; green boxes represent averages ± s.e.m. (**D**) Schematics of the experiments testing the effects of conditioning with isoproterenol (Iso) in adult mice preconditioned with methoxamine pairing (Mtx) the day before. (**E**) Changes in ODI one day after pairing with (After Mtx) and after pairing with Isoproterenol (After Iso). (**F**) Schematics of the experiments testing the effects of conditioning with isoproterenol in mice subjected to long-term MD (LTMD). The eyelids contralateral to the tested hemisphere were sutured at P21 and re-opened after P130 and the visual responses were imaged immediately (Eye open). Following 3 days of normal rearing (Before), after the visual conditioning to the deprived eye (After), and after a day in dark from the visual conditioning (After1d). (**G**) Changes in ODI in LTMD mice induced visual conditioning with isoproterenol. In parenthesis is the number of animals tested at each time point. (**H**) Expansion of panel E detailing the individual changes in ODI induced by the pairing conditioning. Gray box indicates 95% confidential interval of ODI values for the age matched naïve mice.

The online version of this article includes the following source data for figure 5:

**Source data 1.**

responses were imaged again, and after a day in the dark to assess the stability of the induced changes. Mice subjected to LTMD exhibited a very low ODI ($-0.04 \pm 0.02$, n = 14), compared to age matched naïve mice (*Figure 5G*). Seven mice that showed no signs of spontaneous recovery were conditioned with monocular visual conditioning paired with systemic isoproterenol, which resulted in an increased ODI (*Figure 5H*, Before: $-0.01 \pm 0.04$; After: $0.11 \pm 0.05$: p=0.031, n = 7). In four of these mice we confirmed that the changes were maintained after one day in the dark (After1d: ODI: $0.11 \pm 0.04$, n = 4). Altogether the results indicate that the activation of Gs- and Gq-coupled receptors can promote experience-dependent bidirectional changes in ocular dominance in adults in accordance with the pull-push model, and long after the termination of the critical period.

## Discussion

Previously we proposed that the pull-push neuromodulation of Hebbian plasticity by the balance of Gs- and Gq-coupled receptors is a metaplasticity mechanism that allows rapid reconfiguration of the plastic state of cortical synapses over a wide range of possibilities, from LTP-only to LTD-only states (*Huang et al., 2012*). Here we validated the pull-push model in vivo in V1 by demonstrating that manipulations of GPCR signals promote shifts in ocular dominance in the directions predicted by the model. Pairing brief monocular stimulation with the activation of Gq-coupled receptors, predicted to induce an LTD-only state, reduced the cortical responses to the stimulated eye. Conversely, pairing the same monocular stimulation paired with the activation of Gs-coupled receptors, predicted to induce an LTP-only state, enhanced the cortical response to the stimulated eye. The rapid changes induced by these Gq- and Gs-pairings were stable and affected the stimulated eye only, changing ocular dominance in a G-protein specific manner. Importantly, these modifications of cortical responses can be induced in adults, after the end of the critical period, and can ameliorate the imbalance in ocular dominance resulting from prolonged monocular deprivation.

We presented several lines of evidence supporting a role for the pull-push model in cortical plasticity. First, the experiments with applied exogenous agonists and chemogenetics (DREADDs) showing the predicted shifts in ocular dominance indicated that the Gs/Gq-based pull-push metaplasticity can be recruited in vivo. On the other hand, the non-conventional shift in ocular dominance induced by the infusion of a β-adrenergic antagonist during monocular deprivation (see *Figure 4*), is consistent with a role of endogenous Gs/Gq in controlling the polarity of experience-dependent plasticity. This non-conventional decrease of the open eye response, instead of the typical reduction of the closed-eye response, resembles a similar non-conventional shift in ocular dominance reported to occur when the visual cortex is silenced with GABAergic agonists during the MD (*Reiter and Stryker, 1988*). However, in our experiment, the β-adrenergic antagonist (propranolol) did not silence the cortical responses. We surmise that, in agreement with a previous study of us (*Huang et al., 2012*), blocking β-adrenergic receptors was sufficient to shift the endogenous cortical Gs/Gq balance towards Gq enough to bring an LTD-only state. In that scenario, the most active inputs (from the open eye) will be reduced the most. The complementary experiment, inducing an LTP-only state by blocking Gq-GPCRs was not attempted because different Gq-GPCRs, including cholinergic, adrenergic, serotonergic and glutamatergic receptors can substitute each other in enabling LTD (*Choi et al., 2005*). Consequently, achieving an LTP-only state would require the blockade of too many types of receptors, precluding a simple interpretation of the results. Finally, the cortical nature of the changes underlying the neuromodulators-induced ocular dominance shifts is supported by the ex vivo experiments showing that neuromodulator pairings and monocular deprivation do affects a common set of intracortical synapses. Moreover, in the chemogenetics approach to induce ocular dominance shifts, DREADD transfection was restricted to cortex, indicating that neuromodulation of cortical plasticity is sufficient to alter visual responses.

The crucial role of neuromodulators in experience-dependent plasticity in sensory cortices has been well recognized since the initial observations made by *Kasamatsu and Pettigrew, 1976*, *Pettigrew and Kasamatsu, 1978* that in kittens the depletion of catecholamines prevent deprivation-induced ocular dominance shifts, and that superfusion of norepinephrine promote these changes even in adult cats (*Heggelund et al., 1987*; *Imamura and Kasamatsu, 1988*; *Imamura and Kasamatsu, 1991*). Subsequent studies confirmed similar permissive roles for other neuromodulators, including acetylcholine and serotonin (*Bear and Singer, 1986*; *Gu and Singer, 1995*), and also extended these observations to rodent models and to other sensory cortices (*Engineer et al., 2013*;

*Golovin and Ward, 2016*; *Kang et al., 2014*; *Meunier et al., 2017*; *Shepard et al., 2015*; *Weinberger, 2015*). The prevailing interpretation emerging from these studies is that neuromodulators gate cortical plasticity by enhancing sensory perception, and by promoting patterns of cortical activity suitable for plasticity induction via controlling network excitability. This is consistent with a wealth of studies documenting that neuromodulators improve perception in vivo (see *Gelbard-Sagiv et al., 2018*; *Jacob and Nienborg, 2018*; *McBurney-Lin et al., 2019*; *Nadim and Bucher, 2014* for reviews) also affect cellular intrinsic excitability and synaptic inhibition to gate the induction of Hebbian plasticity in vitro (see *Brzosko et al., 2019*; *Palacios-Filardo and Mellor, 2019*; *Pawlak et al., 2010* for reviews). We propose an additional and different mode of action: direct control of the expression, not the induction, of Hebbian plasticity via the Gq/Gs based pull-push metaplasticity mechanism that we previously demonstrated in vitro, in slices. The exact contribution of these three distinct types of neuromodulation to naturally occurring plasticity remains to be determined. This is a challenging task considering that the neuromodulatory systems can interact between them, can substitute each other in promoting plasticity (*Bear and Singer, 1986*), and can stimulate multiple GPCRs, often with opposite actions. One clear example is the case of adrenergic receptors, where the α1- and the β-adrenoreceptors can promote respectively depression or potentiation of the same activated inputs (*Figure 1* and *Figure 1—figure supplement 1*).

The demonstration that the pull-push metaplasticity is operational in vivo has consequences beyond deepening our understanding of how neuromodulators shape cortical plasticity. Pairing monocular stimulation with GPCR agonists could be developed as a tool to bidirectionally manipulate the strength of targeted cortical circuits in a rapid and controlled manner. The neuromodulator pairing elicit stable changes in cortical response comparable in magnitude with those induced by classical deprivation paradigms like MD, but on a faster time scale: hours instead of days (*Lehmann and Löwel, 2008*). Importantly, the neuromodulator approach readily works in post-critical period adults, at an age when the deleterious effects of long-term monocular deprivation, an established animal model of amblyopia, cannot be reversed by simple manipulations of visual experience. The end of the critical period is thought to result from the maturation of mechanisms that constrain the recruitment of Hebbian and homeostatic plasticity, like the strengthening of GABAergic circuitry, for example (reviewed by *Hensch and Quinlan, 2018*; *Jiang et al., 2005*). Consequently, manipulations aimed to restore ocular dominance in adults almost always target the removal of these constrains (*Hensch and Quinlan, 2018*; *Stryker and Löwel, 2018*). In contrast, instead of removing constrains, the targeting of specific GPCRs described here is unique, as it is aimed at selecting the polarity and to increase the gain of the expression of synaptic plasticity. Importantly, several other non-invasive manipulations reported to enhance adult cortical plasticity after the critical period, seemingly do it through other mechanisms. For example, dark exposure (*He et al., 2007*), environmental enrichment (*Sale et al., 2007*), chronic block of serotonin reuptake (*Maya Vetencourt et al., 2008*), locomotion (*Kaneko et al., 2017*), and retinal inactivation (*Duffy et al., 2018*), are thought to rejuvenate cortical plasticity by decreasing the intracortical inhibition, while cross-modal deprivation is thought to reactivate thalamocortical plasticity (*Rodríguez et al., 2018*; *Teichert et al., 2019*). It would be of therapeutical interest to explore complementary synergies between these manipulations and the neuromodulatory pairings in the recovery from long-term monocular deprivation.

## Materials and methods

### Key resources table

| Reagent type (species) or resource | Designation | Source or reference | Identifiers | Additional information |
|---|---|---|---|---|
| Strain, strain background (*Mus musculus*) | C57BL/6J | Jackson Laboratories | RRID:IMSR_JAX:000664 | |
| Strain, strain background (*Mus musculus*) | B6;C3-Tg(Scnn1a-cre)3Aibs/J | Jackson Laboratories | RRID:IMSR_JAX:009613 | |

*Continued on next page*

*Continued*

| Reagent type (species) or resource | Designation | Source or reference | Identifiers | Additional information |
|---|---|---|---|---|
| Strain, strain background (*Mus musculus*) | B6.Cg-Gt(ROSA)26 Sor^tm32(CAG-COP4*H134R/EYFP)Hze/J | Jackson Laboratories | RRID:IMSR_JAX:012569 | |
| Strain, strain background (*Mus musculus*) | ROSA26-LSL-GsDRE ADD-CRE-luc | PMID:28167604 | | |
| Strain, strain background (AAV) | AAV2-CamKIIa-HA -rm3D(Gq)-IRES -mCitrine | UNC Vector Core | | |
| Strain, strain background (AAV) | AAV8-CamKIIa- HA-rm3D(Gs)- IRES-mCitrine | UNC Vector Core | | |
| Strain, strain background (AAV) | rAAV-hSyn-Cre | SignaGen Laboratories | Cat#: SL100888 | |
| Chemical compound, drug | chlorprothixene | Sigma-Aldrich | Cat#: C1671; CAS: 6469-93-8 | |
| Chemical compound, drug | isoflurane | Patterson Veterinary | Cat#: 07-890-8115 | |
| Antibody | anti-GFP antibody, (Chicken, polyclonal) | Aves Labs | Cat#: GFP-1020 RRID:AB_2307313 | 1:2000 |
| Antibody | Alexa 488 conjugated anti-chicken IgY (Goat, monoclonal) | Abcam | Cat#: ab150169 RRID:AB_2636803 | 1:400 |
| Chemical compound, drug | Isoproterenol | Tocris Bioscience | Cat#: 1747 | |
| Chemical compound, drug | Ro 60–0175 | Tocris Bioscience | Cat#: 1854 | |
| Chemical compound, drug | CNO | Enzo Life Sciences | Cat#: BML-NS105-0005 | |
| Chemical compound, drug | propranolol | Sigma-Aldrich | Cat#: P0884 | |
| Chemical compound, drug | nadolol | Sigma-Aldrich | Cat#: N1892 | |
| Chemical compound, drug | methoxamine | Sigma-Aldrich | Cat#: M6524 | |
| Chemical compound, drug | DAPI | Life technologies | Cat#: D-1306 | |
| Commercial assay or kit | Alzet 1007D | DURECT Corporation | Cat#: 0000290 | |
| Software, algorithm | MATLAB | Mathworks | RRID:SCR_001622 | |
| Software, algorithm | Prism | GraphPad Software | RRID:SCR_002798 | |
| Software, algorithm | Igor Pro | Wavemetrics | RRID:SCR_000325 | |
| Software, algorithm | Mini Analysis Software | Synaptosoft | RRID:SCR_002184 | |
| Other | CCD camera | Dalsa | Model#: DS-1A-01M30-12E | |
| Other | ProLong Gold antifade | Life Technologies | Cat#: P36930 | |

## Animals

C57BL/6J [RRID:IMSR_JAX:000664] (Jackson Laboratory, Bar Harbor, ME) mice, Cre dependent Gs-DREADD (ROSA26-LSL-GsDREADD-CRE-luc) (*Akhmedov et al., 2017*); generously gifted from

Rebecca Berdeaux) mice, and layer four specific ChR2 expressing mice (F1 between *Scnn1a*-cre (B6; C3-Tg(Scnn1a-cre)3Aibs/J)[RRID:IMSR_JAX:009613] x Ai32(B6.Cg-Gt(ROSA)26Sor^{tm32(CAG-COP4*H134R/EYFP)Hze}/J)[RRID:IMSR_JAX:012569]; both from Jackson Laboratory) were reared in a 12 hr light/dark cycle. Juvenile mice (P28-35) or fully-grown adult mice (>P130) of both sexes were used. All protocols were approved by the Institutional Animal Care and Use Committee (IACUC) at Johns Hopkins University (protocol # MO17M366) and followed the guidelines established by the Animal Care Act and National Institutes of Health (NIH). Only black Cre dependent Gs-DREADD mice were used for the experiments.

## Optical imaging of the intrinsic signal

Mice were anesthetized with isoflurane (2–3% for induction; 0.7–1.2% for maintenance in oxygen) supplemented with chlorprothixene (2 mg/kg, i.p.) and placed with head-fixed in front of the LCD monitor for visual stimulus. Atropine was injected subcutaneously to reduce mucosal secretion (0.05 mg/kg). Eye drops were administered to keep eyes moist and body temperature was maintained at 37 ˚C with heating pad and rectal probe. Heart rate was monitored throughout the experiment by electrocardiogram. The skull over the V1 region on the left hemisphere was exposed and washed with hydrogen peroxide. Agarose (3%) and an 8 mm round glass coverslip were placed on top of the exposed area. Optical imaging was performed following the method optimized for the measure of the ocular dominance in mice (*Cang et al., 2005*; *Kalatsky and Stryker, 2003*). Briefly, visual response intrinsic signals were acquired using a Dalsa 1M30 CCD camera (Dalsa, Waterloo, Canada) controlled by custom software, ContImage (Continuous Image). The surface vasculature and intrinsic signals were visualized with LED illumination (555 nm and 610 nm, respectively). The camera was focused 600 mm below from the surface of the skull. An additional red filter was interposed to the CCD camera and intrinsic signal images were acquired. A high refresh rate monitor (1024 × 768 @ 120 Hz; ViewSonic, Brea, CA) was placed 25 cm in front of the mouse, with their midline aligned to the midline of the mouse for visual stimulus. The visual stimulus presented was restricted to the binocular visual field (−5˚ to +15˚ azimuth) and consisted of a horizontal thin bar (height = 2˚, width = 20˚) continuously presented for 5 min in upward (90˚) and downward (270˚) directions to each eye separately. The cortical response at the stimulus frequency was extracted by Fourier analysis and the two maps generated with the opposite direction of drifting bar were averaged for each eye to calculate the response amplitude as well as the ocular dominance index (ODI). The ODI was computed as following: (1) the intensity maps were smoothed by 5 × 5 low-pass Gausian filter; (2) the binocular region of interest (ROI) was defined at 30% of peak response amplitude of the smoothed intensity map from the ipsilateral eye; (3) the response amplitude of each eye were calculated by averaging the intensity of all pixels in the ROI; (4) the ODI was calculated by the average of (C-I)/(C+I) of all pixels in the ROI where C and I are the contralateral (C) and ipsilateral (I) eye respectively.

## Monocular visual conditioning

Mice were anesthetized and maintained with isoflurane (2–3% for induction; 1.0–1.5% for maintenance in oxygen) so that the heart rate to be 480–570/min throughout the conditioning session and placed with head-fixed in front of the LCD monitor. Right eye of mice was exposed to visual stimulation consists of black and white drifting gratings phase-reversing at 1 Hz and rotated with step increments of multiples of 22.5˚/min (width 3.72˚; length 71˚; contrast 100%) generated with Matlab (Mathworks, Natick, MA). Visual stimulus was presented in a randomized fashion and lasted for an hour. Neuromodulator agonist or vehicle was injected intraperitoneally 30 min in advance from the start of the visual conditioning. To prevent the abrupt increase of the heart rate by isoproterenol administration, nadolol (5 mg/kg, i.p.) was pre-administrated 20 min before the isoproterenol injection. After the conditioning session, the mouse was kept in a dark environment until being used for the next imaging session.

## Monocular deprivation

Short-term monocular deprivation of juvenile mice (P28-31) in *Figure 4* was done by closing the left eye (ipsilateral from the imaging hemisphere). Long-term monocular deprivation in *Figure 5* was done by closing the right eye (contralateral from the imaging hemisphere) of mice at p21. After the mice anesthetized with isoflurane (2–3% for induction; 1.5% for maintenance), the margins of upper

and lower eye lids were trimmed and sutured shut. Small amount of Neosporin was applied to the sutured eye to prevent an infection. The sutured eye was checked before it was opened for an imaging session to make sure the integrity of the lid suture.

## Osmotic pump drug infusion

Mice receiving subcutaneous drug infusion were implanted with osmotic pumps (Alzet 1007D, DURECT Corporation, Cupertino, CA). The pumps were primed in saline at 37 ˚C. To implant the osmotic pump, mice were anesthetized with isoflurane (2–3% for induction; 1.5% for maintenance) and the pump was installed subcutaneously. Meloxicam (5 mg/kg, s.c.) was administered after the surgery as an analgesic.

## Slice electrophysiology
### Preparation of cortical slices

Brain slices from mice (9–12 weeks) were prepared as described previously with some modifications (*Huang et al., 2012*). Briefly, each mouse was anesthetized using isoflurane vapors, then transcardially perfused with ice cold dissection buffer (dissection buffer in mM: 212.7 sucrose, 5 KCl, 1.25 $NaH_2PO_4$, 10 $MgCl_2$, 0.5 $CaCl_2$, 26 $NaHCO_3$, and 10 dextrose bubbled with 95%$O_2$/5% $CO_2$ (pH 7.4)) and immediately decapitated. The brain was removed and immersed in the ice-cold dissection buffer. Thin (300 μm) coronal slices of visual cortex were cut in the ice-cold dissection buffer and transferred to a light-tight holding chamber with artificial cerebrospinal fluid (ACSF in mM: 119 NaCl, 5 KCl, 1.25 $NaH_2PO_4$, 1 $MgCl_2$, 2 $CaCl_2$, 26 $NaHCO_3$, and 10 dextrose bubbled with 95%$O_2$/5% $CO_2$ (pH 7.4)). The slices were incubated at 30 ˚C for 30 min and then kept at room temperature until they were transferred to recording chamber.

## Whole-cell current clamp recordings for LTP and LTD

Whole-cell current clamp recordings were performed using C57BL/6J mice. Recordings were made from layer 2/3 pyramidal neurons in the V1 with glass pipettes (3–5 Mohm) filled with potassium based internal solution (internal solution in mM: 130 K-gluconate, 10 KCl, 0.2 EGTA, 10 HEPES, 4 Mg-ATP, 0.5 Na-GTP, and 10 Na-phosphocreatine (pH 7.2–7.3, 280–290 mOsm). Excitatory postsynaptic potentials (EPSPs) were evoked every 20 s by two concentric bipolar electrodes (125 mm diameter; FHC, Bowdoin, ME) placed ~0.9 mm apart in the middle of the cortical thickness. Stimulus intensity was adjusted to evoke simple waveform (2–10 mV), short onset latency (<4 ms) monosynaptic EPSPs. Synaptic strength was quantified by measuring the initial slope (the first 2 ms) of EPSPs and input independence was confirmed by the absence of paired-pulse interactions. Mean baseline slope was calculated from 30 consecutive sweeps before the start of CNO (10 μM) application. After 10 min from the perfusion of CNO, Spike-timing dependent plasticity (STDP) was induced by 200 pairings consist of one presynaptic activation given either 10 ms before or 10 ms after four consecutive action potentials (100 Hz) evoked by supra-threshold depolarizing current steps through the recording electrode (~1 nA, 2 ms). After the induction of STDP, CNO was washed out with normal ACSF and the EPSPs were recorded for at least 30 min afterward.

## Whole-cell voltage clamp recordings for pathway specific $Sr^{2+}$ desyncronized EPSCs

Whole-cell voltage clamp recordings were performed using the layer four specific ChR2 expressing mice. Slices were pre-incubated with $Ca^{2+}$-free ACSF containing 4 mM $Sr^{2+}$ and 4 mM $Mg^{2+}$ (95% $O_2$/5% $CO_2$) for 20 min before the initiation of the recordings. To record AMPA receptor mediated desyncronized EPSCs, the recordings were done in the presence of 20 μM bicuculline and 100 μM DL-2-amino-5 phosphonopentanoic acid (DL-APV). Recording pipettes were filled with internal solution containing (in mM): 130 Cs-gluconate, 8 KCl, 1 EGTA, 10 HEPES, 4 ATP, and 5 QX-314, pH 7.4, 285–295 mOsm. Visually identified layer 2/3 pyramidal neurons were voltage clamped at −80 mV. Desyncronized EPSCs were elicited by activating ChR2 from the layer four projections or by stimulating lateral projections with stimulating electrode. ChR2 was activated using 470 nm or white LED illuminated through a 40X objective lens. To stimulate lateral projections, a theta glass stimulating electrode was positioned laterally away about 100–200 μm from the recorded neuron. The minimal

intensity of light (5 ms duration) or electrical stimulation (0.2 ms duration) was determined to elicit a reliable response on a cell-by-cell basis.

Current sweeps were acquired every 10 s for a duration of 1500 ms including a 5 mV test pulse (100 ms duration). A light or electrical stimulation was evoked at 700 ms from the start of each sweep, and a 400 ms window before stimulation was used for quantifying spontaneous desynchronized EPSC events (preStim) and a 400 ms window after a 50 ms delay from stimulation onset was used for quantifying stimulation-evoked desynchronized events (postStim). The desyncronized EPSC event analysis was performed using Mini Analysis software (Synaptosoft, Decatur, GA) with threshold set to three times the root mean square (RMS) noise. Events with rise time >3 ms and cells with an RMS noise >2 were excluded. To calculate the average amplitude of the evoked $Sr^{2+}$ desynchronized EPSC events without spontaneous events, following equation was used: [(postStim amp x postStim freq) - (preStim amp x preStim freq)] / (postStim freq - preStim freq) where amp is amplitude and freq is frequency.

## Viral injection

Adeno-associated virus expressing Gq- or Gs-coupled DREADD under the control of the Ca2+/cal-modulin-dependent protein kinase II (CamKII) promoter was injected to the V1 region of 3 weeks old mice bilaterally (Gq-coupled DREADD: AAV2-CamKIIa-HA-rm3D(Gq)-IRES-mCitrine; Gs-coupled DREADD: AAV8-CamKIIa-HA-rm3D(Gs)-IRES-mCitrine; from UNC Vector Core, Chapel Hill, NC). C57BL/6J mice of either sex were anesthetized and head fixed in a stereotaxic device (Kopf Instruments, Los Angeles, CA) under 1.5–2% isoflurane. The superficial layer of the V1 was targeted with following coordinate: bregma −3.6, lateral 2.5, depth 0.3. AAVs (800 nl) were injected with syringe infusion pump. After the injection, mice were recovered on a heat pad and were returned to their home cage once they awake.

Viral injection of the recombinant adeno-associated virus expressing Cre recombinase under the control of human synapsin promoter (rAAV-hSyn-Cre; from SignaGen Laboratories, Rockville, MD) was done by bulk regional viral injection to the visual cortical area of neonatal Cre dependent Gs-DREADD mice at p1. Briefly, neonatal mice were cryo-anesthetized (*Phifer and Terry, 1986*) in an ice-cold chamber and positioned dorsal side up secured with an adhesive bandage across the upper body. The visual cortical area was guided with anatomical landmarks, including occipital fontanelle and lambdoid suture, which are visible through the neonatal skin. 50 nl of rAAVs were injected right beneath the skull slowly with syringe infusion pump (5 nl/sec). After the injection, the mice were kept on a heat pad for the recovery and returned to their home cage once awake.

## Immunohistochemistry

The expression of Gs-DREADDs in the V1 was examined by visualizing the GFP expression conjugated to the DREADDs at the end of the optical imaging experiments. The GFP signal from these mice were hardly detectable, so the GFP fluorescence signal was magnified with fluorochrome conjugated antibody. The anesthetized mice were transcardially perfused with 10% neutral buffered formalin solution. Following perfusion, the brain was extracted and kept in the fixative solution overnight. The brain was sliced into 70 µm coronal sections and the slices were transferred to phosphate buffered saline (PBS). The slices were permeabilized in 0.2% Triton-X 100 in PBS solution for an hour, blocked in a 5% normal goat serum for an hour, and washed in PBS. The slices were then treated overnight at 4 ˚C with chicken anti-GFP antibody (1:2000)[RRID:AB_2307313] (Aves Labs, Inc, Davis, CA). Then the slices were washed in PBS and treated with Alexa 488 conjugated goat anti-chicken IgY antibody (1:400)[RRID:AB_2636803] (Abcam, Cambridge, UK) for 2 hr. The slices were washed in PBS for an hour and mounted on a slide glass. Slides were coverslipped with the Prolong Gold anti-fade mounting solution with DAPI (Cell Signaling Technology, Inc, Danvers, MA) incorporated.

## Drug administration

Isoproterenol (Tocris Bioscience, Bristol, UK), Ro 60–0175 (Tocris), propranolol (Sigma-Aldrich, St. Louis, MO), nadolol (Sigma-Aldrich) and methoxamine (Sigma-Aldrich) were prepared in saline. CNO (Enzo Life Sciences, Farmingdale, NY) was dissolved in 5% DMSO and 95% saline.

## Statistical analysis

For two sample comparisons we used Matlab to perform the Wilcoxon signed-rank test, in the case of paired samples, and the Wilcoxon rank sum, in the case of non-paired samples. For comparison of multiple groups, we used Prism (GraphPad Software, San Diego, CA) first to confirm the normal distribution of each group with the Shapiro-Wilk test. Then we performed a one-way ANOVA followed by Holm-Sidak *post hoc* test for the comparison of independent groups or repeated measure of ANOVA followed by *post hoc* Bonferroni's multiple comparisons test for the comparison of paired groups. Data are presented as averages ± s.e.m.

## Acknowledgements

This research was supported by the National Eye Institute of the National Institutes of Health under grant number R01EY012124 (to AK). We thank Rebecca Berdeaux for facilitating the ROSA26-LSL-GsDREADD-CRE-luc mice and Dr. Hey-Kyoung Lee for helpful discussions and comments.

## Additional information

### Funding

| Funder | Grant reference number | Author |
|---|---|---|
| National Eye Institute | R01EY012124 | Alfredo Kirkwood |

The funders had no role in study design, data collection and interpretation, or the decision to submit the work for publication.

### Author contributions

Su Z Hong, Conceptualization, Investigation, Visualization, Methodology, Writing - original draft; Shiyong Huang, Daniel Severin, Investigation; Alfredo Kirkwood, Conceptualization, Resources, Funding acquisition, Writing - review and editing

### Author ORCIDs

Alfredo Kirkwood  https://orcid.org/0000-0001-9148-9742

### Ethics

Animal experimentation: This study was performed in strict accordance with the recommendations in the Guide for the Care and Use of Laboratory Animals of the National Institutes of Health. All of the animals were handled according to approved institutional animal care and use committee (IACUC) protocols of Johns Hopkins University. The protocol was approved by the Committee on the Ethics of Animal Experiments of Johns Hopkins University (protocol#: MO17366). All surgery was performed under isofluorene anesthesia, and every effort was made to minimize suffering.

### Decision letter and Author response

Decision letter https://doi.org/10.7554/eLife.54455.sa1
Author response https://doi.org/10.7554/eLife.54455.sa2

## Additional files

### Supplementary files

• Transparent reporting form

### Data availability

All data generated or analysed during this study are included in the manuscript and supporting files.

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
