## [Decision Letter]

**Acceptance summary:**

Your previous work showed that neuromodulators can enhance synaptic potentiation or depression via alterations in neuronal and circuit excitability. This paper is an important advance, demonstrating that these neuromodulatory events regulate ocular dominance plasticity in vivo. It is especially exciting that the neuromodulation can influence ocular dominance well after the juvenile critical period normally associated with ocular dominance development, and that driving these neuromodulatory circuits can rescue function after visual deprivation.

**Decision letter after peer review:**

Thank you for submitting your article "Pull-push neuromodulation of cortical plasticity enables rapid bi-directional shifts in ocular dominance" for consideration by *eLife*. Your article has been reviewed by three peer reviewers, one of whom is a member of our Board of Reviewing Editors, and the evaluation has been overseen by Gary Westbrook as the Senior Editor. The following individuals involved in review of your submission have agreed to reveal their identity: Carlos D Aizenman (Reviewer #2).

The reviewers have discussed the reviews with one another and the Reviewing Editor has drafted this decision to help you prepare a revised submission.

Summary:

This is an interesting study linking neuromodulation of synaptic plasticity by Gq/Gs G-proteins to ocular dominance plasticity in vivo. The authors show in vivo using chemogenetic stimulation that modulating synapses in the visual cortex either with a Gq11 or a Gs coupled DREADD determines the direction of evoked synaptic plasticity. Remarkably, even in adult mice, reactivating Gs/Gq could reinstate plasticity even after the close of the critical period. The paper is an advance over previous work on this topic, and provides results of interest to the field.

Essential revisions:

1) A better diagram for Figure 1 would help.

2) Figure 2: Were the behavioral tests done at very different times? The targeting of the hSyn virus vs. the CaMKII virus could be different (infection of all neurons vs. pyramidal cells). Do you have any data to show expression?

3) It would be interesting to speculate whether any of the other manipulations reported to enhance adult cortical plasticity after the critical period could work via this same mechanism?

4) It would be interesting to see whether spontaneous minis changed in Figure 3, to distinguish between pathways specific effects and generalized decreases (like synaptic scaling). If the authors collected spontaneous activity they could easily do this comparison. I don't think new experiments are needed, but if they have the data, why not check?

5) The authors mention bidirectional plasticity, however, in their experimental design, they only evoke plasticity in one direction, using activation of either Gq or Gs coupled pathways. Do the authors predict that sequential activation of Gq followed by Gs at some delayed interval (or vice versa), could display true bidirectional plasticity in the same set of synapses? Along these lines, in Figure 5, a ceiling effect was invoked to explain the lack of increased ODI by isoproterenol in adults, and this was tested by decreasing ODI via prolonged monocular deprivation, which could then be increased by visual stimulation paired with isoproterenol. Might an alternative had been to first decrease ODI by pairing visual conditioning with methoxamine, then attempt to increase ODI by pairing visual stimulation with B-adrenergic activation at some interval later to truly show metaplasticity gated by GPCRs at the same sets of synapses in vivo?

---

## [Author Response]

Essential revisions:1) A better diagram for Figure 1 would help.

We have changed the diagram in Figure 1A to better illustrate the conditioning and measurements.

2) Figure 2: Were the behavioral tests done at very different times? The targeting of the hSyn virus vs. the CaMKII virus could be different (infection of all neurons vs. pyramidal cells). Do you have any data to show expression?

Yes, the Gq-DDREAD was tested earlier in mice 4 weeks older than those for Gs-DDREAD. We did those experiments first, before we learned how to transfect new-born pups. Although we tried, the CaMKII virus for the Gs-DREADDs never gave good levels of expression to test effects on ODI. That is why we used the other approach (hSyn). Please note that the question answered by these experiments was whether a local (non-systemic) activation of Gs/Gq is sufficient to shift the ODI. Defining the cell types involved in the DREADD-assisted shifts in ODI is an interesting question, but beyond the scope of this study. In consideration of these issues we added a brief disclaimer in the Results section:

“Although we cannot rule out that the different transfection approaches might have enhanced or moderated the opposite effects obtained with the Gq and Gs-DREADDs, the simplest explanation for the outcome is that the DDREADs affected plasticity as predicted by the pull-push idea.”

As for the images, we did not collect them systematically as it was not our intention to analyze them. In general, we made a visual confirmation that the visual cortex was expressing the DREADDs. We archived a few images like the one presented in Author response image 1 but only for the Gs DREADD. We kept a few Gq DREADD images from the slice experiments.

**Author response image 1. respfig1:** Fluorescence images of V1 area from the Gs-DREADDs mouse injected with rAAV-hSyn-Cre. Slices were immunostained for GFP to visualize the Gs-DREADDs expressing neurons. Dotted lined squares indicate the region magnified at the bottom. Scale bar, 200 µm.

3) It would be interesting to speculate whether any of the other manipulations reported to enhance adult cortical plasticity after the critical period could work via this same mechanism?

We have expanded the last paragraph of the Discussion to state:

“The end of the critical period is thought to result from the maturation of mechanisms that constrain the recruitment of Hebbian and homeostatic plasticity, like the strengthening of GABAergic circuitry, for example (reviewed by Hensch and Quinlan, 2018; Jiang et al., 2005). Consequently, manipulations aimed to restore ocular dominance in adults almost always target the removal of these constrains (Hensch and Quinlan, 2018; Stryker and Löwel, 2018). In contrast, instead of removing constrains, the targeting of specific GPCRs described here is unique, as it is aimed at selecting the polarity and to increase the gain of the expression of synaptic plasticity.”

4) It would be interesting to see whether spontaneous minis changed in Figure 3, to distinguish between pathways specific effects and generalized decreases (like synaptic scaling). If the authors collected spontaneous activity they could easily do this comparison. I don't think new experiments are needed, but if they have the data, why not check?

We did measure the spontaneous minis. In the experiments with Ro 60-0175 the spontaneous minis recorded in the conditioned (C) cortex were not significantly smaller than those recorded in the non-conditioned (NC) cortex (see Author response image 2).

NC: 18.546 + 0.918pA (n=20)

C: 17.028 + 0.788pA (n=21)

(rank sum, p=0.309)

**Author response image 2. respfig2:** 

Although these results argue against the idea of a generalized scaling-like decrease, we would prefer not to report them because spontaneous minis are difficult to interpret. They are a mixture of events caused by spontaneous and evoked release, and all that confounded by the presence of Sr+2. Besides, the fact that with Ro 60-0175 only layer IVlayer II/III pathway was affected, which by itself argues against a generalized scaling-like decrease.

5) The authors mention bidirectional plasticity, however, in their experimental design, they only evoke plasticity in one direction, using activation of either Gq or Gs coupled pathways. Do the authors predict that sequential activation of Gq followed by Gs at some delayed interval (or vice versa), could display true bidirectional plasticity in the same set of synapses? Along these lines, in Figure 5, a ceiling effect was invoked to explain the lack of increased ODI by isoproterenol in adults, and this was tested by decreasing ODI via prolonged monocular deprivation, which could then be increased by visual stimulation paired with isoproterenol. Might an alternative had been to first decrease ODI by pairing visual conditioning with methoxamine, then attempt to increase ODI by pairing visual stimulation with B-adrenergic activation at some interval later to truly show metaplasticity gated by GPCRs at the same sets of synapses in vivo?

That was a very appropriate suggestion to test the ceiling effect in the adults. The experiments worked: after reducing the ODI with methoxamine conditioning, the conditioning with isoproterenol increased the ODI. The results are presented in Figure 5D, E.